# Widely tunable and narrow-linewidth violet lasers enabled by UV-transparent materials

C. A. A. Franken [1]✉, W. A. P. M. Hendriks[2,3], L. V. Winkler[1,4], A. R. do Nascimento Jr [5], A. van Rees[1], M. Dijkstra[2], S. Mardani[2], D. Kienzler[6,7], R. Dekker[8], J. van Kerkhof[5], P. J. M. van der Slot[1,9], S. M. García-Blanco[2,3] & K.-J. Boller[1]

Embedding multi-wavelength lasers in photonic waveguide circuits is of interest for next-generation ion traps, such as for miniaturizing optical clocks or upscaling ion-based quantum computing. Critically, this path involves photonic integration of highly coherent lasers in the ultraviolet (UV) range, which is presently obstructed by the transparency limit of materials used in established integrated waveguides. Here, we demonstrate the first integrated, extended cavity diode laser based solely on UV-transparent materials. We integrate aluminum oxide waveguide circuits with gallium nitride amplifiers to generate milliwatt-level on-chip output power near the ultraviolet range. The extended cavity approach allows for wide wavelength coverage and precise frequency control, which is demonstrated by tuning mode-hop-free to a Sr-transition frequency. Due to the inherent stability of photonic circuits and UV-compatible integration, the intrinsic laser linewidth reaches a record-low value around 300 kHz with better than 43-dB side-mode suppression. These results announce the viability of a novel class of integrated lasers that opens access to the UV.

On-chip generation of highly coherent light with MHz-level frequency control is attractive for many integrated photonic applications. These reach from miniaturizing ion traps for optical clocks achieving improved portability[1,2] to upscaling ion-based quantum computers[3,4]. As a follow-up to bulk ion traps[5], photonic integration of beam delivery to microfabricated surface traps introduces an intrinsic sub-wavelength stability of optical paths at a compact form factor. Examples of integrated traps showed coherent addressing of individual strontium (Sr⁺) qubit ions at a 674-nm transition[6] and trapping of ytterbium (Yb⁺) ions using 435-nm radiation[7]. Spectral expansion to multi-wavelength routing enabled coherent control of Sr⁺ ions confined in a surface trap[8].

So far, all photonic integration for ion traps relied on the silicon nitride ($Si_3N_4$) waveguide platform, which is a leading platform for the infrared and visible range[9,10]. However, most ions demand much shorter ultraviolet (UV) wavelengths, for ionization and laser cooling, and for readout of quantum states. Examples are Ca⁺ ions at 393 and 397 nm[11] or Yb⁺ ions at 370 nm[12] corresponding to photon energies of 3.1 to 3.4 eV. Some ions lie at the edge of the UV (Sr⁺ at 408 nm, Yb⁺ at 411), other ions of interest require even deep-UV radiation such as beryllium and magnesium at 313 and 280 nm (4.0 and 4.4 eV, respectively). At ultraviolet photon energies, silicon nitride cannot be used due to material-intrinsic propagation loss, even if fabrication is optimized for weak scattering[13].

¹Laser Physics and Nonlinear Optics, Department of Science and Technology, MESA+ Institute of Nanotechnology, University of Twente, Enschede, The Netherlands. ²Integrated Optical Systems, Department of Science and Technology, MESA+ Institute of Nanotechnology, University of Twente, Enschede, The Netherlands. ³Aluvia Photonics B.V., Enschede, The Netherlands. ⁴TOPTICA Photonics SE, Gräfelfing, Germany. ⁵PHIX B.V., Enschede, The Netherlands. ⁶Department of Physics, ETH Zurich, Zurich, Switzerland. ⁷Quantum Center, ETH Zurich, Zurich, Switzerland. ⁸LioniX International B.V., Enschede, The Netherlands. ⁹Nonlinear Nanophotonics, Department of Science and Technology, MESA+ Institute of Nanotechnology, University of Twente, Enschede, The Netherlands. ✉e-mail: c.a.a.franken@utwente.nl

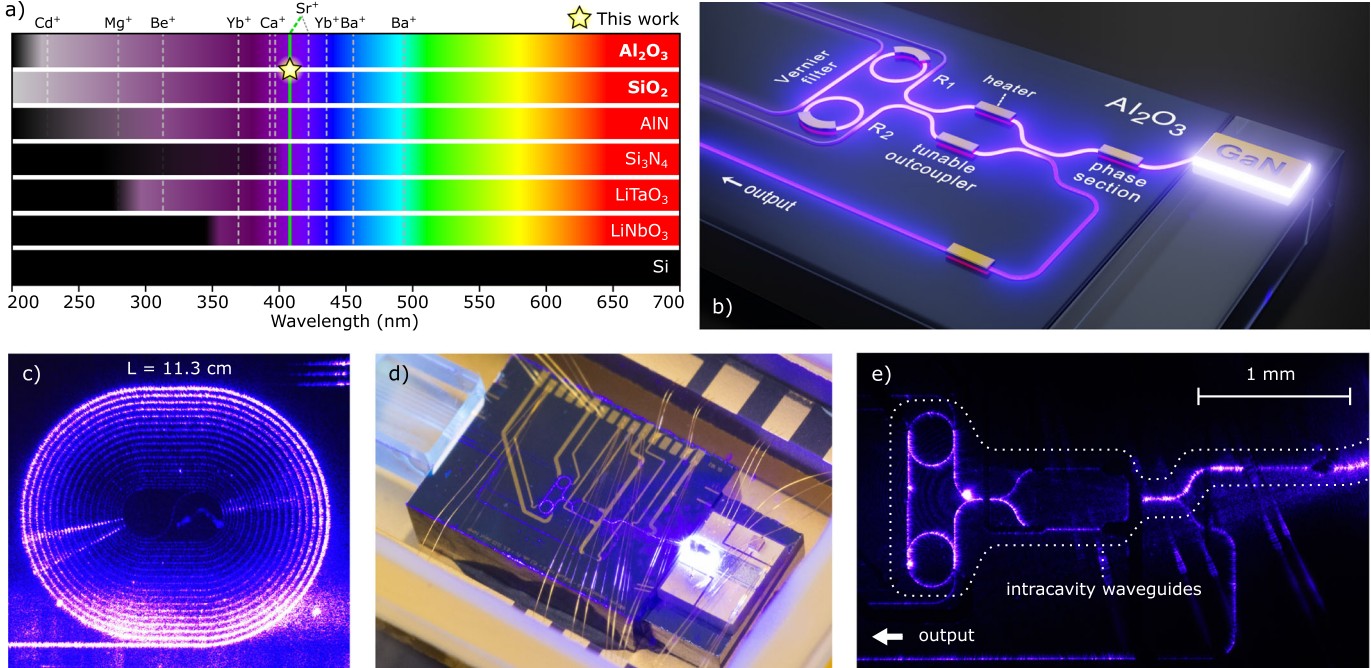

**Fig. 1 | Highly coherent, integrated violet lasers using UV-transparent materials. a** Coarse overview of the transparency of selected photonic materials. Several key ion transitions at UV and blue wavelengths, such as for quantum computing and optical clock applications, are denoted by vertical lines. The laser presented in this work combines the wide transparencies of the higher-index $Al_2O_3$ core material with a $SiO_2$ cladding and targets a $Sr^+$-ion transition wavelength around 408 nm. As a measure for the transparency limit we choose a value of 10 dB/cm for the film propagation loss (where available) as a practical lower bound, and the reported material bandgap values as the absolute transparency limit (opaque black shading). Material data references: $Al_2O_3$[14], $SiO_2$[16,73], AlN[14], $Si_3N_4$[13], $LiTaO_3$[74], $LiNbO_3$[75], Si[76,77]. **b** Laser design using a double-pass gallium nitride (GaN) optical amplifier edge-coupled to an aluminum oxide feedback chip. Light from the amplifier travels into the feedback waveguide circuit and is diverted toward a pair of sequentially arranged micro-ring resonators ($R_1$, $R_2$) of slightly different radii forming a spectrally narrowband Vernier filter. Only light that is resonant with both ring resonators can travel back into the GaN chip for laser amplification. Thin-film metal heaters allow for wavelength tuning, fine alignment of the laser cavity length, and for adjusting the laser output coupling. **c** As a key factor in the laser design, the propagation loss of the waveguides was measured by injecting light at 405 nm in a series of spirals. Transmission measurements of these structures with various length reveal a propagation loss of $\alpha = 2.8 \pm 0.3$ dB/cm. **d** Using a UV-compatible integration method, the GaN amplifier and $Al_2O_3$ feedback chip are hybrid integrated and packaged in a hermetically sealed nitrogen environment. **e** Close-up picture of the $Al_2O_3$-based laser in operation, showing scattered light from intracavity waveguides.

Unlike silicon nitride, making use of aluminum oxide ($Al_2O_3$) as core material with a silicon dioxide ($SiO_2$) cladding for an integrated waveguide platform would be an excellent candidate for the complete range of wavelengths required by trapped ions. This can be seen in Fig. 1a which gives an overview of the transparency ranges of various relevant materials. The huge transparency range of $Al_2O_3$ reaches beyond the deep-UV, due to its wide bandgap near 7.6 eV or 165 nm[14,15], while $SiO_2$ shows an even wider bandgap of 9.3 eV or 133 nm[16]. Propagation losses lower than 3 dB/cm and micro-ring resonators with quality factors up to 1.5 million have so far been reported in the violet and UV ranges, between 360 and 405 nm[14,17,18]. Compared to propagation loss values of 60.4 dB/cm for AlN waveguides at 379 nm[19], most recent measurements showed low 0.4-dB/cm film loss and 1.3-dB/cm waveguide propagation loss at the same wavelength in $Al_2O_3$[20]. Such low loss values suggest that aluminum oxide integrated circuits can extend routing of light to trapped ions deep into the ultraviolet, while conditioning of light with modulators may rely on piezo-optomechanical implementations[21] or platform-independent techniques[22,23].

However, providing the full benefit of integration in terms of size, efficiency and stability would have to include chip-integrated UV light sources. Most promising for such source integration are semiconductor optical amplifiers coupled with aluminum oxide waveguide circuits forming chip-sized extended cavity diode lasers, exhibiting tunable and narrow-linewidth emission.

Early work on chip-size, narrow linewidth lasers started with semiconductor feedback circuits, intended to address the infrared telecom range, an approach where small material bandgaps are sufficient, such as offered by Si waveguides (1.1 eV) or InP waveguides (1.3 eV). The situation changed dramatically when high-contrast, low-loss waveguides with a much wider bandgap became available, such as in $Si_3N_4$. This reduced both linear and nonlinear material losses, which enabled widely tunable and highly frequency selective waveguide circuits for maximizing laser coherence with extended on-chip photon lifetimes[24–26]. Recently, the wide transparency of $Si_3N_4$ also enabled the first realization of hybrid integrated lasers in the visible range, with mW-level fiber-coupled output at red wavelengths[27,28]. To reach shorter wavelengths, feedback chips were so far only used for self-injection locking and frequency pulling of Fabry-Pérot diode lasers near their main oscillating mode, using manually aligned, laser-external feedback from single ring resonators. Milliwatt-level fiber-coupled powers were obtained in the blue range (450-460 nm) and about 500 µW in the violet[29]. However, at the very end of the visible spectrum, tuning remained fundamentally limited, with linewidths wider than several-MHz, and powers restricted to values between 10 and 100 µW[30,31], respectively. With the silicon nitride platform, reaching out to shorter wavelengths in the UV range can be excluded[13,32], for example when using 5-eV GaN diodes at 250 nm[33]. This would be quite different when employing UV-transparent aluminum oxide as waveguide platform material. However, lasers employing aluminum oxide have so far only been using off-chip pumped, rare-earth ion doped waveguides operating at much longer near and mid-infrared wavelengths[34,35], where propagation loss is less of a concern.

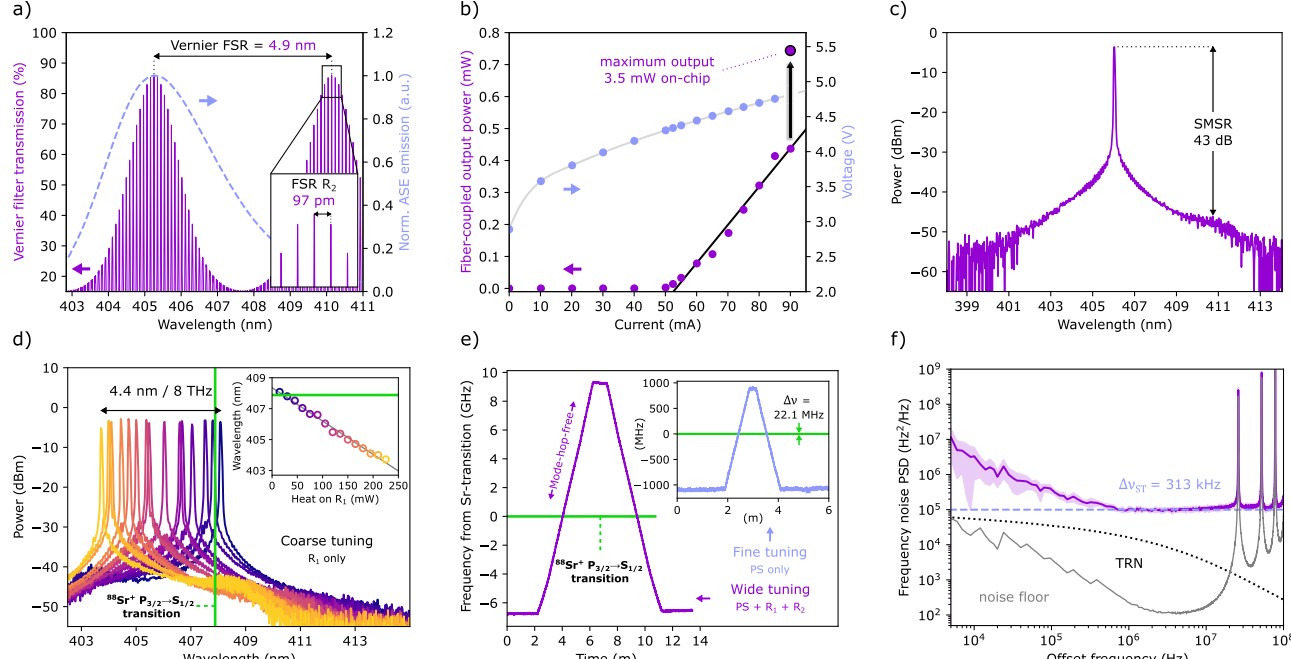

**Fig. 2 | Laser characteristics and performance. a** Calculated transmission function of the Vernier filter (solid line). The dashed trace is the normalized spectrum of amplified spontaneous emission (ASE) of the solitary amplifier. **b** Fiber-coupled output power versus drive current, showing a laser threshold and slope efficiency of 50 mA and 12 μW/mA, respectively. The maximum fiber-coupled output power was $0.74 \pm 0.04$ mW measured at 405.5 nm, corresponding to about 3.5 mW ($\pm 50\%$) on-chip power in the output waveguide. The secondary axis shows the corresponding IV-curve for the GaN amplifier driving the laser. **c** Optical spectrum showing single wavelength operation with a high side-mode suppression ratio (SMSR) of 43 dB, at an amplifier current of 90 mA. **d** Superimposed laser spectra recorded with various heater settings of the first ring resonator ($R_1$). The laser tunes in wavelength more than 4.4 nm (or 8 THz) from 408.1 nm to 403.7 nm. **e** Precise control over the laser is shown by mode-hop-free (MHF) tuning the laser across the $^{88}Sr^+ P_{3/2} \rightarrow S_{1/2}$ transition frequency ($v = 734.99068$ THz or $\lambda \approx 408$ nm). Wide tuning (16 GHz) is enabled by serially tuning the phase section (PS) and both ring resonator ($R_1$ and $R_2$) heaters and finer tuning is enabled by only adjusting the phase section heater. Separate measurements showed a continued operation at the transition frequency for more than 70 minutes and a maximum of 21.5 GHz of MHF tuning at other wavelengths (see Supplementary Information). **f** Measured single-sided power spectral density (PSD) of the laser frequency noise (purple line) using delayed self-heterodyne detection. The spectrum shows a white noise level of 100 $\pm 8$ kHz²/Hz; corresponding to an intrinsic laser linewidth of 313 $\pm 25$ kHz, which is at least an order of magnitude above the calculated, laser cavity thermorefractive noise (TRN, black dotted line).

Here we present the first extended cavity diode laser with feedback waveguides based solely on UV-transparent materials (Fig. 1b). The laser operates at a wavelength immediately neighboring the UV, at the very end of the visible range. To provide low-loss waveguides we use aluminum oxide cladded with silicon dioxide. The extended cavity laser is formed by composing $Al_2O_3$ waveguide building blocks into a tunable feedback circuit and by UV-compatible integration with a GaN amplifier, providing high frequency stability, wide tunability and a high on-chip output power. Pursuing this concept further in the UV is of interest for quantum computing and optical clocks, while our laser is already of direct interest for $Sr^+$ ion traps. Our approach substantiates the viability of integrated, fully tunable and coherent light sources in the UV range where amplifier materials are under development[36,37], leveraged by the wide transparency range of aluminum oxide.

## Fabrication, design and integration

Aiming on gain-wide tunability and narrow intrinsic linewidth using the novel fabrication method, we devise frequency selective feedback from a waveguide circuit containing a Vernier filter comprising of two micro-ring resonators with slightly different radii of 150 and 153 μm, respectively (Fig. 1b)[38]. Using measurements and numerical simulations, a 400-nm wide and 100-nm thick $Al_2O_3$ waveguide core embedded in $SiO_2$ was chosen to provide single-transverse-mode propagation ($TE_{00}$) and reduced sidewall scattering while maintaining tight guiding in curved waveguides. The fabrication using, for the first time, sputtering of nano-crystalline $Al_2O_3$-films for short wavelengths in the UV, employs CMOS-compatible techniques on a wafer scale (details in Methods). The sputtered film in combination with chemical-

mechanical polishing and an ultra-high quality oxide cladding, results in lower scatter and propagation losses. The nano-crystalline structure allows for annealing temperatures above the crystallization temperature of amorphous $Al_2O_3$ ALD films (>800 °C[14,39]), which greatly improves the quality of the deposited oxide and reduces the propagation loss.

With a mask design for a 10-cm diameter wafer a total of 127 chips are fabricated. The wafer carries, next to a variety of laser feedback circuits, a large set of test structures which are the building blocks that constitute the feedback circuit. The latter are waveguide tapers for mode conversion, thermo-optic phase shifters, micro-ring resonators, tunable couplers consisting of a Mach-Zehnder interferometer and directional couplers. In addition, weakly curved waveguide spirals of various lengths are fabricated for measurements of propagation loss (Fig. 1c). Characterizing individual components allows to identify suitable Vernier feedback circuits for hybrid integration with GaN amplifiers within a single wafer run.

Employing aluminum oxide waveguides for novel feedback circuits operating near the UV requires considering various critical circuit functions. One is providing tunability of the optical filtering to realize wavelength tuning of the laser. For tuning across the entire amplifier bandwidth we choose a free spectral range of the Vernier filter as wide as the amplified spontaneous emission spectrum (ASE, Fig. 2a). Single wavelength output with high side-mode suppression requires the Vernier filter transmission to be high and sufficiently narrowband compared to the laser's mode spacing. Fulfilling these conditions simultaneously asks for a functional circuit design with all parameters properly set. These are the diameters of the micro-ring resonators; the

degree of power coupling between bus and ring waveguides; the shape of tapers for mode matching between the amplifier, feedback chips and fibers; and the amount of output coupling from the laser resonator.

A parameter of central importance is the waveguide loss, as it determines the laser threshold and frequency selectivity of the feedback circuit. For quantifying the propagation loss, we perform single-pass transmission measurements, using a set of waveguide spirals with weak curvature and different lengths (Fig. 1c). The measurements yield a propagation loss of $\alpha = 2.8 \pm 0.3$ dB/cm at a wavelength of 405 nm (details in Supplementary Information), on par with fabrication via more delicate deposition processes like ALD[14]. The Vernier filters that were selected for integration with amplifiers have ring resonators with a large FSR (FSR of 99 and 97 pm respectively) and strong bus-to-ring coupling. The selection is motivated by offering narrow spectral filtering, while still providing high transmission and feedback to the amplifier which is essential for reaching laser threshold and high on-chip power (Fig. 2a).

The suite of building blocks contains chip-to-chip couplers, here, lateral tapers that are designed for the feedback circuit waveguide mode matching the mode field for the selected GaN amplifier, achieving a much increased 31% coupling efficiency compared to earlier work[30] (amplifier details in Methods). Fresnel reflections coupling back into the laser mode at the facets are suppressed by letting the waveguides form angles with regard to their facet normals, with the angles calculated using Fresnel's refraction law and the effective mode indices. Tapers are also designed for efficient coupling into output fibers. Actively controllable building blocks, as part of our platform development, are an intracavity phase shifter for cavity mode alignment with the Vernier filter; two phase shifters for fine-tuning the optical length of the ring resonators; an outcoupler with a tunable coupling range between 10% and 90%; and a phase section in the output waveguide. Tuning these circuit components is realized via the thermo-optic effect through metal thin-film heaters, with a measured phase-shifting efficiency of 175 mW/π.

The overall design of the laser also needs to provide long-term passive frequency stability. Manual alignment of edge-coupled DFB laser diodes and feedback chips, with mechanical stages, is sufficient for spectral narrowing of intrinsic linewidths to the Hertz-level[40], however, acoustic perturbations and thermal drift typically limit stability to time scales shorter than milliseconds. In contrast, long-term frequency and power stability, and inherent robustness to external perturbations, require mutual bonding of chips, hermetic sealing and temperature stabilization[41]. With infrared lasers, the laser cavity mode is allowed to propagate through bonding materials, and hermetic sealing is not essential. Both have been demonstrated with hybrid integration[42] and heterogeneous integration[32]. However, at the much higher UV photon energies, bonding materials need to be kept out of the laser mode and hermetic sealing is indispensable for long-term stable output. In this work, two lasers were hybrid integrated with epoxy-free optical paths at chip-to-chip and chip-to-fiber interfaces, and packaged with hermetic sealing in a butterfly diode laser housing for maximum UV compatibility (see Fig. 1d, more details in Methods).

## Results

Brightly lit intracavity waveguides show successful operation of the laser when driven above threshold (see Fig. 1e). For increasing pump current we measure the fiber-coupled output power after optimization of the heaters of the ring resonators, the phase section and the tunable outcoupler. We find a laser threshold current of 50 mA, above which the output power increases approximately linearly with pump current (see Fig. 2b). Remaining deviations from the linear trend can arise from a combination of factors like thermal cross-talk between heaters, slight hysteresis from the gain medium when changing a heater voltage and our use of a digital-to-analog voltage source to drive the heaters

causing a limited resolution. In order to present long-term stable power values, these points are measured in the weeks after first operation, when the GaN amplifier is burnt-in[43] and the output power has settled. We note that, directly after packaging of the laser, we find a maximum, fiber-coupled optical power of 0.74 ± 0.04 mW at a wavelength of 405.5 nm. Correcting for losses at the chip-to-fiber coupling interface (6.7 ± 2.1 dB, see Supplementary Information), the laser generates about 3.5 mW (± 50%) on-chip power in its output waveguide, which is the applicable value for fully on-chip integrated ion traps considered here as application. We note that the fiber-coupled output levels near the UV are between 4 and 40-times higher than what has been reported recently with self-injection of Fabry-Pérot lasers using silicon nitride resonators with a manual, stage-coupled set-up[30,31]. Possible reasons for higher output can be better mode matching between amplifier and feedback chip or our tunable control of the laser outcoupling.

Wide-range near-the-UV output spectra are recorded with an optical spectrum analyzer. An emission spectrum of the laser shows single-wavelength operation with a high side-mode suppression ratio of 43 dB (Fig. 2c), which is more than two orders of magnitude higher than previously reported around this wavelength range in silicon nitride[30,31]. The recording may indicate a full-width-at-half-maximum laser linewidth below the resolution limit of the optical spectrum analyzer (90 GHz or 50 pm). For exploring the wavelength coverage we vary the heater power only at the first ring resonator ($R_1$) which tunes the laser wavelength in steps of the free spectral range of the other resonator ($R_2$). Figure 2d shows superimposed laser spectra with a linear tuning response as a function of heating power on the first ring resonator (see inset). The minor power fluctuations and deviations from equidistant wavelength tuning are attributed to the same mechanisms that cause deviations from the linear trend in the output power curve (Fig. 2b), as discussed in the previous paragraph. The record-wide wavelength coverage amounts to 4.4 nm (8 THz), corresponding to tuning over the entire ASE spectrum of the amplifier.

Coarse tuning allows for a fast and full wavelength access within the wide spectral coverage of the laser, however, this is achieved through mode hops of the laser. In contrast, achieving smooth, gapless tuning to a precisely predefined, fixed optical frequency of choice within the gain bandwidth is crucial for locking the laser to narrow atomic transitions of trapped ions. Figure 2e illustrates mode-hop-free tuning of the laser over a continuous, 16 GHz-wide span that includes the optical frequency of 734.99068 THz ($\approx$ 408 nm) of the $P^{3/2} \rightarrow S^{1/2}$ transition of $^{88}Sr^+$ ions, measured using a calibrated, high-resolution wavelength meter. Wide mode-hop-free tuning is accomplished by serially adjusting the phase section and both ring resonator heaters in a fixed 5.2:1:1 heating power ratio, while finer, high-resolution sweeps can readily be achieved by varying only the phase section heater (see inset). Separate experiments showed that this triple-heater tuning scheme supports up to 21.5 GHz of hop-free tuning at other wavelengths, and the laser can remain within the Sr-transition linewidth for almost the entire recording of more than 70 minutes (see Supplementary Information).

To verify the promise of passive stability inherent to fully integrated lasers[44], we investigate the laser frequency stability on various timescales. A recording using a high-resolution wavelength meter shows that the laser was oscillating mode-hop free for at least 84 minutes, and that the average drift is low, around 1 MHz/min in the last 24 minutes of recording (see Supplementary Information). We attribute this long-term stability to integration of the chips and as well to hermetic sealing and packaging, which includes thermal stabilization. For faster frequency fluctuations, beneath technical laser noise and above the cavity thermorefractive noise, the linewidth is ultimately limited by spontaneous emission of photons into the cavity mode. This limit, also called the intrinsic linewidth, depends mainly on the quality and length of the laser cavity. For a detailed description see Methods.

We note that such measurement is not straight forward in this spectral range and self-heterodyne measurements of these integrated lasers have not been performed before. Here, we record the power spectral density (PSD) of the laser frequency noise using delayed self-heterodyne detection, see Fig. 2f. The purple line shows the laser frequency noise averaged over the calculated PSD of 10 separate traces of 1-ms duration, with the standard deviation denoted by the shaded area. The arm-length difference used for detection is about 7.6 m (~ 11.4 m optical length), which is consistent with the appearance of peaks in the noise floor at 26 MHz and integer multiples. Technical noise can be seen to decrease continuously with increasing offset frequency up to about 700 kHz where the noise levels-off to constant power density. This white noise component is the signature of having reached the quantum limit (Schawlow–Townes limit) associated with random phase fluctuations through photons spontaneously emitted into the laser mode[26,45]. The average white noise level represented by the blue line corresponds to a linewidth of $313 \pm 25$ kHz, which is a record-low value for integrated diode lasers in this spectral range. This result is in alignment with the theoretically expected, quantum-limited linewidth (>63 kHz, see Methods) and well above the calculated, laser cavity thermorefractive noise (see Methods).

## Discussion

We have shown the first chip-integrated extended cavity diode laser solely using UV-transparent materials. Specifically, we use feedback waveguides made of aluminum oxide and silicon dioxide. The high performance of the laser is made possible through important advances such as: low losses, UV-compatible integration, optimized laser design and advanced laser control. The laser in this work operating at 405 nm outperforms previous work on integrated lasers close to this wavelength on every metric and, secondly, is expandable to deeper UV wavelengths free from material absorption. The output power is 4 to 40-times higher, and the side-mode is suppression one to two orders of magnitude higher than with other platforms[29–31]. The spectral linewidth of around 300 kHz is at least an order of magnitude lower than in previous work, noting that only one other work provided an upper estimate of several MHz[30]. The laser frequency noise is well above the calculated, laser cavity thermorefractive noise which solidifies that we measured the quantum-limited linewidth of the laser. Another source of noise can be autofluorescence, which may be observed in visible and UV-transparent waveguides at high detection sensitivity. However, at the wavelengths relevant here, autofluorescence in $Al_2O_3$ was found to be one-to-two orders of magnitude weaker than in $Si_3N_4$ waveguides[46], noting that the latter waveguides are successfully employed already in single-photon sensitive applications[6]. Reaching lower intrinsic linewidths for aluminum oxide based extended cavity lasers can be achieved with lower propagation losses and longer cavity lengths. We also find excellent longer-term passive frequency stability, which we address to our UV-compatible mutual integration of gain and feedback waveguides, followed by packaging and sealing. The 8-THz coarse laser tuning range is on par or exceeds any integrated laser within the entire visible and infra-red range regardless of the used waveguide platform[26,27,30,47,48]. Finally, we demonstrate a mode-hop-free tuning range more than an order of magnitude wider than previous work[30].

The aluminum oxide waveguides fabricated with a novel, CMOS compatible sputtering technique show low propagation loss. While absorption loss should play an insignificant role in this wavelength range due to the high material bandgap of aluminum oxide (7.6 eV), we address the present waveguide loss mainly to sidewall scattering. This can be concluded from observing much lower loss in thin films of 0.4–0.6 dB/cm at 377 nm[20,49]. Reduction in sidewall scattering can be achieved by using a higher-selectivity mask which should reduce shadowing, optimization of the etching recipe and multi-pass exposure during the lithography step[50]. This work has delivered a variety of functional waveguide building blocks allowing for precise dynamic

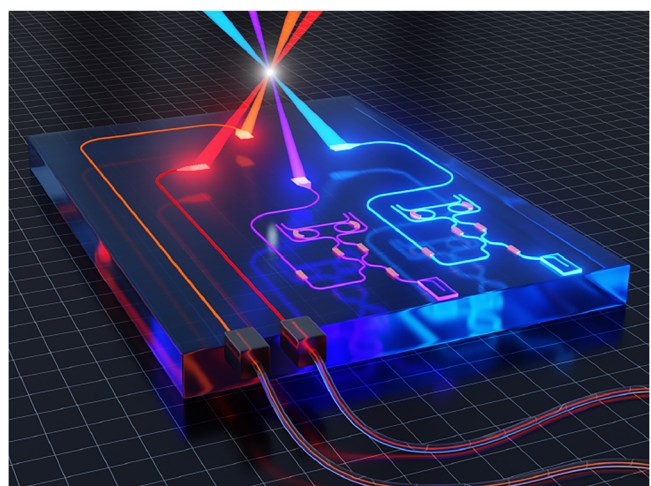

**Fig. 3 | Future embedding of on-chip lasers with integrated ion traps.** Vision for embedding on-chip, extended cavity diode lasers directly in the photonic beam delivery of a generic, integrated ion trap. Using aluminum oxide ($Al_2O_3$) as a waveguide platform enables multi-wavelength generation and operation spanning from the infrared into the UV. Alternatively, light for interrogating the ion can be brought on-chip by fiber. Trap electrodes and light conditioning elements are omitted for more clarity.

control of light in the circuit. The merging with active semiconductor waveguide components, yielding high-performance lasers to work as on-chip light engines, opens a viable path for integrated photonics into the UV. Along the path towards full integration of multi-wavelength lasers with ion traps using a single waveguide chip, intermediate steps for portable systems can be taken as required. These may also involve fiber transport from separate lasers such as envisioned in Fig. 3.

We believe that the innovations in design and implementation of UV-capable waveguide circuits and light sources presented here will have their most immediate impact on lasers specifically refined to drive transitions of selected ions, such as, the $^{88}Sr^+$ $P_{3/2} \rightarrow S_{1/2}$ transition at 408 nm. The coarse and mode-hop-free tuning measurements demonstrated here to address the frequency of the named transition are enabled by the broad wavelength coverage of these lasers and their precise MHz-level controllability and thus pave the way towards adoption for Sr+ and also other ion traps. For cooling and state preparation[51], the demonstrated laser fulfills the typical power, tuning range and linewidth requirements, which can make such lasers central ingredients for miniaturizing portable optical clocks such as for space-based applications[52]. In the realm of quantum computing using ions, cryogenic cooling is not seen as a strict requirement because laser Doppler cooling still keeps the ions at mK temperatures[51]. The cooling transition of the $^{40}Ca^+$ ion is as close as a few nanometers to the tuning range of our laser at 397 nm[53]. Shifting the center wavelength of the gain by several nanometers can be straightforwardly obtained via a slightly different stoichiometry when growing GaN-type amplifiers[54]. With typical requirements of less than 100 μW of optical cooling power at the ion and given the long-standing availability of efficient (> 50%) grating couplers for beam delivery[55], our on-chip laser power is already well within the power specification for cooling multiple ions. Due to the high passive stability of the laser, with an average drift of only 1 MHz/min, already a slow stabilization to keep the laser on resonance with the transition can be performed using the on-chip heaters with the ions as the reference. For transitions with long lifetimes, fast frequency stabilization might be required, like the $^{171}Yb^+$ $D_{5/2}$ transition at 411 nm[56]. Stabilization and fast control of the temporal structure, beyond the typical millisecond limit of thermal actuators[23] used here, may either be achieved with direct current modulation of the gain element, piezo-optomechanical implementations[21] or with back-

routing to active elements[57]. Similarly, the on-chip available power can be increased with multiple amplifiers[58,59] whereas attaining high powers in the UV by nonlinear upconversion has only recently reached mW-level powers[60]. Given the intrinsic UV-transparency, universal laser concept and advancing material development for GaN amplifiers, our method can be engineered for further development into the UV range, possibly also into the deep-UV[18].

## Methods

### Feedback chip design

To identify a suitable core cross-section for fabrication, preceding measurements were consulted. In the UV at 377 nm, for a 170-nm thick $Al_2O_3$ film deposited with a reactive sputtering process and followed by chemical mechanical polishing, we measured slab propagation losses of $0.6 \pm 0.3$ dB/cm[49]. Selecting a thinner waveguide would reduce propagation loss via reduced sidewall scattering while the necessity for tight guiding of UV wavelengths puts a lower bound for the waveguide thickness. Simulation of optical modes (2D finite element methods, Lumerical) in straight and bended waveguides, and integrals of mode intensity at the sidewalls were performed to select an $Al_2O_3$ core cross-section of 400-nm wide by 100-nm tall. To account for fabrication tolerances and variation in amplifier performance the total roundtrip losses in the laser cavity can be controlled by selecting a feedback circuit with appropriate coupling strength for the rings, determined by the bus-to-ring directional coupler. Elements in the circuit such as the phase section, outcoupler and ring resonators can be tuned in phase over $3.5\pi$ using the on-chip heaters. All heater adjustments can be made, manually or automatically, using a low-noise, multichannel power supply (Tunable Laser Controller, Chilas B.V.). Varying the outcoupling allows for maximizing the laser output at each filter setting and amplifier current. Characterization of tunable test couplers (see Supplementary Information) show that the out-coupling of the laser can be adjusted between approximately 10% and 90% with a heater-based, phase-shifting efficiency of 175 mW/$\pi$.

### Fabrication

The $Al_2O_3$ waveguide fabrication is carried out by the Integrated Optical Systems group in the Nanolab cleanroom of the MESA+ Institute (University of Twente), which starts by depositing a 110-nm thick $Al_2O_3$ layer, using an optimized RF reactive sputter deposition process[61], onto an 8-$\mu$m thick thermally oxidized 10-cm diameter silicon wafer. A chemical mechanical polishing step is used to reduce the surface roughness of the deposited $Al_2O_3$ layer from ~1.5 nm to less than 0.2 nm RMS[20], and reducing the layer thickness to the targeted 100 nm. The waveguides are patterned using e-beam lithography and etched with reactive ion etching. The resulting waveguides are fully buried by an 8-$\mu$m thick, high temperature and high quality LPCVD $SiO_2$ cladding and subsequently annealed at 1150 ℃[20]. To implement thermo-optic tuning on the feedback chip, resistive heaters are fabricated using a lift-off process, by deposition of a 10/10 nm Cr/Pt layer for the resistive heaters and topped with a 300-nm thick Au layer for low-resistance leads.

### Hybrid integration and packaging

The gain of the laser is provided by an InGaN/GaN double-pass amplifier (superluminescent diode, SLED, from Exalos AG) at a center wavelength of 405 nm and emitting into a single transverse mode at a polarization matching that of the $Al_2O_3$ waveguides ($TE_{00}$). The diode is highly-reflective coated (reflectivity vs. air > 95%) on its back facet and anti-reflective coated (reflectivity vs. air < 0.1%) on the facet facing the $Al_2O_3$ chip. The amplifier has a mode-field diameter of 1.87 by 0.6 $\mu$m (width and height). The GaN waveguide towards the output facet is angled at 9.2 degrees with respect to the facet normal. Together with the AR coating this eliminates Fresnel back reflections into the guided mode to the $10^{-9}$ level[62], which prevents locking of the laser

to that facet[63]. During the integration process, the amplifier is aligned and edge-coupled to the $Al_2O_3$ feedback chip. Here, we verify that the fundamental transverse mode is launched into the $Al_2O_3$ waveguide and the coupling loss is minimized, by maximizing brightness of light scattered within the feedback circuit. Using Lumerical we calculate a theoretical coupling efficiency of 91% of the GaN amplifier mode with the $Al_2O_3$ mode, with less than a reduction of 7%-point coupling efficiency for a 100-nm misalignment (within specification of the used mechanical stages). When optimum alignment is reached, the chips are mutually integrated by bonding using a UV curable epoxy, keeping the optical path free of it. On the output side of the $Al_2O_3$ chip the facets are polished at 8 degree angles (out-of-plane), to prevent back reflections into the lasing mode, and an array of polarization maintaining, APC fibers is attached. Packaging of the laser comprises electric wire bonding, including a thermistor and Peltier element for temperature control of the laser, set to 20 ℃ for all measurements. One laser is packaged in a nitrogen atmosphere sealed with a glass lid for visual inspection of the circuit in operation, such as seen in Fig. 1d and e. The measurements are carried out with a metal seamwelded lid variant to provide improved hermetic sealing in an argon atmosphere for extending the lifetime[64].

### Experimental details

For the mode-hop-free tuning experiments the laser frequency is recorded using a calibrated HighFinesse WS6-200 wavelength meter. The long-term drift of the laser frequency, just after laser startup, is recorded using a HighFinesse WS-U 1645 wavelength meter. The linewidth of the laser is below the resolution of the available optical spectrum analyzer (Ando AQ6315A). To measure the intrinsic laser linewidth delayed self-heterodyne experiments, using existing methods[65], are carried out with a fiber delay length of 7.6 m (-11.4 m optical length), an acousto-optic modulator (G&H FiberQ, model S-M200-0.4C2A-3-F2P) and a high bandwidth avalanche photodiode (Thorlabs APD430A2), with signals recorded using an oscilloscope (Agilent MSO6104a). To match the polarization of the signal from both arms on the photodiode, a fiber polarization controller is used in one of the arms. A total of 10 traces with a 1-ms duration is recorded. Each trace is subsequently analyzed to obtain the frequency noise power spectral density, with first performing a Hilbert transform on the signal to retrieve the instantaneous phase. After unwrapping the phase the first order time derivative is calculated to obtain the instantaneous frequency, from which the carrier frequency (here, 200 MHz) is subtracted. Finally, to obtain the frequency noise power spectral density the instantaneous frequency is Fourier transformed using different window sizes and averaged (more details in the Supplementary Information). The noise floor is obtained by recording a frequency noise spectrum with balanced arms, i.e., zero delay.

### Calculation of intrinsic laser linewidth

We use a theoretical approximation of the linewidth as derived in previous work[26] and adapt it to the specifics of the laser presented here. The adapted, Schawlow-Townes laser linewidth is described as[66] (Eq. (1))

$$\Delta\nu_{ST} = \frac{1}{4\pi} \cdot \frac{\nu_g^2 h\nu \, n_{sp}\gamma_m\gamma_{tot}(1+\alpha_H^2)}{P_{out,tc}\eta_P} \cdot \frac{\alpha_P}{F^2}. \tag{1}$$

A detailed description of each parameter and derivation is described in the Supplementary Information. In eq. (1), $\nu_g = c/n_{g,gain}$ is the group velocity in the gain section with $c$ the speed of light and $n_{g,gain}$ the group index, $h\nu$ is the photon energy using $\nu = c/\lambda_0$ with $\lambda_0$ the vacuum wavelength. $n_{sp}$ is the population inversion factor, i.e., the ratio between the spontaneous rate of downward transitions and the stimulated rate of downward and upward transitions[67,68]. $\alpha_H$ is Henry's linewidth enhancement factor, describing the strength of the gain-

index coupling in the gain section[69]. $\gamma_m$ and $\gamma_{tot}$ describe the distributed mirror and total cavity loss coefficients, respectively. Here, $\gamma_{tot}$, also includes the frequency dependent loss of the Vernier filter. The type of lasers discussed here have various output channels, however, for this particular case the waveguide after the tunable outcoupler denoted by *output* in Fig. 1b is used to measure the laser output power. The previous description of the Schawlow-Townes linewidth[26] uses the output power measured at the back facet of the amplifier waveguide. The adapted description we use here[66], converts the on-chip power directly after the tunable coupler $P_{out,tc}$ to the power at the back facet of the amplifier waveguide by multiplying $P_{out,tc}$ with a conversion factor $\eta_P$. Here, to obtain $P_{out,tc}$, we take the on-chip power at the end facet $P_{out}$ and backpropagate it to right after the outcoupler using the propagation loss. The gain of the laser has to compensate for the large and asymmetric outcoupling in the cavity, which adds extra phase noise due to the amplified spontaneous emission coupling into the laser mode, expressed by the Petermann factor $\alpha_P$[70]. The factor $F$ describes linewidth narrowing[26], via cavity extension and by frequency dependent loss (or filtering) of the $Al_2O_3$ Vernier-filter circuit. The calculated linewidth is lowest when detuning the cavity mode into the low-frequency wing of the Vernier peak, as was done during the experiments through optimizing the phase section for lowest linewidth. Here, eq. (1) predicts a linewidth of 63 kHz.

### Calculation of the thermorefractive noise

Previous work has shown that thermorefractive noise can be a dominant thermal noise source in the frequency stability of micro-ring resonators[71,72]. To calculate the thermorefractive noise of our full laser cavity (detailed description in Supplementary Information), we first recognize the separate components that constitute the laser cavity, which consists of three sub-resonators. These are two micro-ring resonators (Vernier filter) and another cavity extension formed by the bus waveguides. As a second step, we evaluate the thermorefractive noise $S_{\delta f_{R1}}$ of ring resonator 1 using a finite element method (FEM) simulation in COMSOL as described by Huang et al.[71]. The thermorefractive noise for the other two cavity contributions, i.e., ring resonator 2 and the bus waveguide sub-resonator, can be obtained by scaling $S_{\delta f_{R1}}$ with the ratio of the cavity mode volume to the mode volume of ring resonator 1, see Supplementary Fig. 4 in the Supplementary Information. After having introduced the various cavity components, we note that the thermorefractive noise of the full laser cavity cannot be treated via a single volume in which the noise has the same effect on the laser frequency noise uniformly along the laser cavity length. The reason is that laser wavelength changes are different for phase changes in the bus waveguides compared to phase changes in the ring resonators[42]. To take this into account, we describe the full cavity thermorefractive noise as

$$S_{\delta f_{laser}} = f'^2_{R1} S_{\delta\theta_{R1}} + f'^2_{R2} S_{\delta\theta_{R2}} + f'^2_{bus} S_{\delta\theta_{bus}} \tag{2}$$

Here, the amplitude pre-factors $f'_i$ are obtained numerically by finding the shift of the resonance frequency of the laser cavity $\delta f_{laser}$ mode when changing the phase $\delta\theta_i$ in the respective sub-resonator. The thermorefractive phase noise of each sub-resonator is given by

$$S_{\delta\theta i} = \frac{4 S_{\delta f_i}}{\kappa^2_{c,i}} \tag{3}$$

Here, $\kappa_{c,i}$ is the effective linewidth of each sub-resonator, which can be numerically found from experimental data (Supplementary Tables 1 and 2 in the Supplementary Information). The calculated thermorefractive noise, as shown in Fig. 2f, is below the measured laser frequency noise by at least 1 order of magnitude (at 1-MHz offset frequency), and the measured noise floor is even lower in the region of

interest. This signifies that the measured linewidth at high offset frequencies is given by the Schawlow-Townes limit and not the thermorefractive noise.

## Data availability

Data supporting the plots and other findings are available from the corresponding author upon request.

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

## Acknowledgements

We would like to thank Okay Canti and Christian Nölleke (TOPTICA Photonics) for arranging lab equipment that was used for some of the laser characterization measurements, Florian Sterl (Sterltech Optics) for illustrating several figures, and Gijs van Ouwerkerk (PHIX) for the photograph shown in Fig. 1d. Guanhao Huang (Harvard University) for the discussion and help with simulating the thermorefractive noise. All authors were funded by the following grants for their work on this project: Dutch Research Council (NWO) under the grant "Ultra-narrowband lasers on a chip" (16718) and Rijksdienst voor Ondernemend Nederland Holland High Tech through High-Tech Systems and Materials (HTSM) (PPS_2020_90).

## Author contributions

C.A.A.F. established the development of the laser. C.A.A.F., L.V.W., A.v.R. contributed to the feedback circuit design. W.A.P.M.H., M.D, S.M., S.M.G.-B. covered the various aspects of development and fabrication of the Al$_2$O$_3$ photonic chips, with support of C.A.A.F. The hybrid integration was performed by A.R.d.N., with contributions from R.D. and J.v.K. The experiments were carried out by C.A.A.F. L.V.W. supported the long-term frequency stability measurements and A.v.R., L.V.W. carried out linewidth measurements and initial analysis. C.A.A.F., K.-J.B., D.K., P.J.M.v.d.S. contributed to discussions regarding ion-traps and lasers. The manuscript was prepared by C.A.A.F., K.-J.B. and edited by all authors.

## Competing interests

Authors W.A.P.M.H. and S.M.G.-B. co-founded Aluvia Photonics B.V. to commercialize the fabrication of Al$_2$O$_3$ photonic chips. The remaining authors declare no competing interests.
