## [Transparent Peer Review file · Nature Communications]

Widely tunable and narrow-linewidth violet lasers enabled by UV transparent materials

Corresponding Author: Dr Cornelis Franken

Version 0:

Reviewer comments:

Reviewer #1

(Remarks to the Author)

The editors asked me to let them know if my previous concerns have been addressed by the revisions. Although the authors have improved certain technical aspects of their resubmitted version of the article, it must be noted that my main comments have not been taken into account.

The benefit of using a material potentially relevant for deep-UV photonic integrated circuits and laser sources has still not been demonstrated as the current wavelength of the laser source is in the visible spectral region.

The authors still report no applications with the current laser source, which would nevertheless be welcome for an interdisciplinary journal. They only suggest speculative applications, now with 88Sr^+ or Ca^+ ions, which might be fair for a specialized optical journal. Considering that there is no experimental result about integrated ion trap in the paper, the relevance of Fig.1 a) is still questionable for a scientific journal.

Interestingly, the new version provides an estimation of the impact of the conventional thermo-refractive noise, which excludes this contribution as the source of excess noise beyond 700 kHz. I take advantage of this new report to suggest to the authors another possible source of noise, namely the autofluorescence of the alumina. There are indeed few papers reporting autofluorescence in alumina, which could be more concerning in the deep-UV wavelength range. Note that an in-depth study of the UV autofluorescence of the alumina is necessary to validate this material for quantum or sensing applications. In this regard, the paper might be overly enthusiastic in the suggested applications.

To summarize, my conclusions regarding the novelty and impact are the same as my previous report. The paper is an interesting intermediate step towards a breakthrough result, that's why it is more appropriate for a specialized optical journal.

Reviewer #2

(Remarks to the Author)

While much work has been done to integrate SOAs with external cavities at telecom wavelengths to both SiN and Si waveguide based PICs, this work appears to be the first in the blue/UV regime.

In this work the authors integrate (by butt-coupling and adhesive attach) a blue light (403-408nm) double pass SOA with an alumina waveguide platform that is intrinsically transparent and low loss from the deep UV to near IR. The alumina PIC has a Vernier ring reflector to create an extended cavity and thermal tuners to provide wavelength tuning, an architecture that has been used effectively at telecom wavelengths by other authors.

To my knowledge this is the first report of an extended cavity blue wavelength laser integrated with a PIC with a deep UV compatible waveguide materials set. This work extends features that have been shown at 1550 nm into new materials system of GaN emitters and alumina waveguide platforms. I believe this work is novel and would be of interest to both the quantum and photonics community at large.

The methods and approach by the authors is sound and meets standards of our field.

A couple of questions for the Authors:

1. I presume that the SLED was first packaged and wire bonded to enable it to be powered to enable an active alignment to the PIC. The authors mention that optimal alignment was done by looking for the maximizing the brightness of scattered light in the reflector. Can the authors add a sentence or two to clarify the steps of the hybridization between the SLED and PIC.

2. The GaN emitter has a relatively small mode size of $1.87 \times 0.6 \mu\text{m}$, most work of this type quantifies or models the optical loss at the coupling interface due to modal overlap with the spot size converter, and the associated coupling loss due to lateral or vertical misalignment in the the two interfaces.

3. It might be of interest to the community to see the impact to optical loss in the alumina if it were not annealed at high temperature. Ion traps often have integrated metal electrodes for trapping of the ions, the presence of buried metals often precludes the use of high temperature process steps for annealing the oxides and alumina film. (or provide a reference to this information)

Reviewer #3

(Remarks to the Author)

The manuscript presents a significant step forward in integrated photonics with the demonstration of a widely tunable and narrow-linewidth laser operating near the ultraviolet range. The use of aluminum oxide waveguide circuits and gallium nitride amplifiers is well-justified and innovative. The reported intrinsic linewidth of 300 kHz and the high output power achieved are particularly impressive and demonstrate the potential for enabling applications like quantum computing and optical clocks. This work introduces a laser that operates at the very edge of the UV spectrum, showing promise for extending photonics platforms into shorter wavelengths.

While the work is technically strong, the claims of deep-UV capability seem a bit overstated since the laser operates in the violet region, specifically 403–408 nm. The title and abstract could be adjusted to better align with the demonstrated results to avoid potential confusion for readers. The novelty of the platform is evident, particularly in the use of aluminum oxide for UV-compatible feedback circuits, but the manuscript could benefit from a clearer distinction between incremental and groundbreaking contributions.

The connection of this laser to quantum technology is promising, but the manuscript could elaborate further on how the laser's characteristics specifically meet the needs of quantum computing or other applications. The authors mention ion traps and cooling transitions, but to fully support the claims of usefulness in quantum applications, additional experimental validation would be very helpful. For example, mode-hop-free tuning should be characterized to determine how far the laser can be tuned continuously, as this is a critical parameter for many quantum systems. Even better, a demonstration of the laser being used in a real quantum experiment would significantly strengthen the claims. This could be as simple as a spectroscopy measurement that directly showcases the laser's applicability in a quantum setting. Without such demonstrations, the claims about potential applications risk appearing speculative and lack convincing evidence of practical usefulness.

It's a great piece of work, but it falls short in showing the laser in action within a real system. Including some system-level measurements would not only convince the audience of the laser's practicality but would also elevate the paper's impact significantly. For instance, integrating the laser into a testbed setup, such as for quantum spectroscopy or state preparation of ions, would add immense value. These demonstrations would provide tangible proof of the laser's capabilities and make the work far more compelling for both academic and industrial audiences.

Overall, this work is a significant engineering achievement, and the authors have demonstrated strong expertise in photonics. The manuscript is well-written and presents solid results that will be of interest to the photonics community. Addressing the points mentioned above would strengthen the manuscript and make its contributions clearer to a broad audience. With some revisions and demonstrations of the laser's performance in a practical setting, this work has the potential to make a valuable and lasting impact on the field of integrated photonics and its applications in quantum technologies.

Version 1:

Reviewer comments:

Reviewer #1

(Remarks to the Author)

The current manuscript presents high-quality work that advances the state of the art in terms of technical performance. The use of a UV-transparent platform is a noteworthy step; however, it appears to be an intermediate milestone rather than a definitive breakthrough toward the ultimate goal of realizing an integrated UV laser source.

While the reported results are promising, they fall short of providing a scientifically rigorous demonstration that the proposed materials will indeed enable UV-integrated laser operation. Furthermore, the manuscript lacks a concrete implementation in

a real-world application where a UV laser is specifically required, which would strengthen the case for its practical relevance.

Additionally, the title remains misleading. It implies that widely tunable and narrow-linewidth violet lasers were achieved as a direct consequence of employing a UV-transparent material. This is inaccurate, as similar violet laser performance has already been demonstrated using silicon nitride—a material not classified as UV-transparent (see Ref. 30: doi: 10.1038/s41566-022-01120-w).

In summary, the revised version does not adequately address the core concerns previously raised. As such, I believe the manuscript still falls short of the standards expected for publication in a multidisciplinary, high-impact journal.

Reviewer #2

(Remarks to the Author)

Thankyou to the authors for satisfactorily addressing the main points I outlined in the previous draft. I have no further comments regarding the revision.

Reviewer #3

(Remarks to the Author)

Thank you for your effort in addressing all the comments.

July 31, 2025

We thank the reviewers for their time and effort. In the following we give our point-by-point response to questions and comments, and point to according changes to the manuscript. The reviewer comments are shown in black and our answers in blue. Changes made to the main text, figure captions, methods and supplementary information are highlighted in blue as well.

Regarding your comments, we are glad having received them as they clearly helped us to revise the manuscript in a couple of important aspects. These are:

- Demonstrate mode-hop-free tuning to the fixed frequency of a strontium ion transition
- Update the linewidth model and calculations
- Better sorting of breakthrough vs incremental progress in the introduction and discussion

Reviewer #1 (Remarks to the Author):

The editors asked me to let them know if my previous concerns have been addressed by the revisions. Although the authors have improved certain technical aspects of their resubmitted version of the article, it must be noted that my main comments have not been taken into account.

The benefit of using a material potentially relevant for deep-UV photonic integrated circuits and laser sources has still not been demonstrated as the current wavelength of the laser source is in the visible spectral region.

We believe it is important to perceive the first-of-its-kind demonstration of an Al_2O_3 -GaN integrated laser as opening a viable future for novel UV sources. We think that providing this vision is much more important. The vision is based on realizing a laser that is exclusively based on deep-UV transparent materials, making it relevant and inspiring for corresponding applications. We note that the UV potential of Al_2O_3 waveguides is already recognized, and researchers start to wonder whether a tunable and coherent light source can directly be integrated as well [doi.org/10.1038/s41467-022-31989-8]. For further clarification related to your question, we want to refer to our response to reviewer #3 where we address the deep-UV claim.

The authors still report no applications with the current laser source, which would nevertheless be welcome for an interdisciplinary journal. They only suggest speculative applications, now with 88-Sr^+ or Ca^+ ions, which might be fair for a specialized optical journal. Considering that there is no experimental result about integrated ion trap in the paper, the relevance of Fig.1 a) is still questionable for a scientific journal.

For balancing the impression that applications of the investigated type of lasers are only “speculative”, also motivated by comments of another reviewer, we have carried out additional fine wavelength tuning experiments with our laser source.

We show tuning of the laser to the frequency of 734.99068 THz (near 408 nm) marked by a calibrated, high quality wavelength meter (absolute 200 MHz, resolution 4 MHz), which is the $P^{3/2} - S^{1/2}$ transition frequency of 88-Sr^+ ions. Tuning is shown to be mode-hop-free, across a wide

span of 21.5 GHz and is finely resolving as required for ion trap applications. Even right after laser start-up the thermal drift, without stabilization, is as low as 1 MHz/minute which allows to keep the laser even with manual corrections at the target frequency. During a 70+ minute recording the laser frequency stays mostly within the 22.1 MHz linewidth of the Sr-transition with minor, manual adjustments of a single thermo-optic phase shifter (see Supplementary Information Fig. SI-2c).

We have modified the manuscript and Supplementary Information to include these measurements, such as in Fig. 2 and Fig. SI-2.

Interestingly, the new version provides an estimation of the impact of the conventional thermo-refractive noise, which excludes this contribution as the source of excess noise beyond 700 kHz.

Indeed, with the present choice of circuit parameters, thermo-refractive noise beyond 700 kHz offset frequency is almost one to three orders of magnitude lower than the intrinsic (Schawlow-Townes) linewidth.

I take advantage of this new report to suggest to the authors another possible source of noise, namely the autofluorescence of the alumina. There are indeed few papers reporting autofluorescence in alumina, which could be more concerning in the deep-UV wavelength range. Note that an in-depth study of the UV autofluorescence of the alumina is necessary to validate this material for quantum or sensing applications. In this regard, the paper might be overly enthusiastic in the suggested applications.

Based on concerns that Al_2O_3 waveguides might suffer from autofluorescence (AF), the reviewer notes it would be necessary to carry out an in-depth investigation of AF for validating the platform for quantum and sensing applications. We agree that an in-depth research of the AF properties of Al_2O_3 waveguides is a highly welcome ingredient for an emerging UV platform. However, we note that the suitability of Al_2O_3 waveguides can already be implied with the present state of the art, by comparing to Si_3N_4 waveguides that are already used in quantum applications. Most recently, in the wavelength range where our laser operates, it has been reported that AF in Al_2O_3 waveguides is one-to-two orders of magnitude weaker than in Si_3N_4 waveguides [doi.org/10.1364/OL.551736]. And such Si_3N_4 waveguides are being successfully employed to integrate ion traps [doi.org/10.1038/nnano.2016.139], where even single AF photons may cause dephasing events. A hint of AF in Al_2O_3 being negligible also in the UV is that such waveguides have been employed around 360 nm for structured-illumination microscopy where weak AF from biological species is recorded without any noticed background from Al_2O_3 [doi.org/10.1038/s41467-022-31989-8]. Based on these results the authors see Al_2O_3 as promising for application also in the deep UV.

To convey this information, we have added the following sentence to the manuscript discussion section:

“Another source of noise can be autofluorescence, which may be observed in visible and UV-transparent waveguides at high detection sensitivity. However, at the wavelengths relevant here, autofluorescence in Al_2O_3 was found to be one-to-two orders of magnitude weaker than in Si_3N_4 waveguides [doi.org/10.1364/OL.551736], noting that the latter waveguides are successfully employed already in single-photon sensitive applications [doi.org/10.1038/nnano.2016.139].”

To summarize, my conclusions regarding the novelty and impact are the same as my previous report. The paper is an interesting intermediate step towards a breakthrough result, that's why it

is more appropriate for a specialized optical journal.

Reviewer #2 (Remarks to the Author):

While much work has been done to integrate SOAs with external cavities at telecom wavelengths to both SiN and Si waveguide based PICs, this work appears to be the first in the blue/UV regime.

In this work the authors integrate (by butt-coupling and adhesive attach) a blue light (403-408nm) double pass SOA with an alumina waveguide platform that is intrinsically transparent and low loss from the deep UV to near IR. The alumina PIC has a Vernier ring reflector to create an extended cavity and thermal tuners to provide wavelength tuning, an architecture that has been used effectively at telecom wavelengths by other authors.

To my knowledge this is the first report of an extended cavity blue wavelength laser integrated with a PIC with a deep UV compatible waveguide materials set. This work extends features that have been shown at 1550 nm into a new materials system of GaN emitters and alumina waveguide platforms. I believe this work is novel and would be of interest to both the quantum and photonics community at large.

The methods and approach by the authors is sound and meets standards of our field.

We thank the reviewer for the comments and proper description of our work.

A couple of questions for the Authors:

1. I presume that the SLED was first packaged and wire bonded to enable it to be powered to enable an active alignment to the PIC. The authors mention that optimal alignment was done by looking for the maximizing the brightness of scattered light in the reflector. Can the authors add a sentence or two to clarify the steps of the hybridization between the SLED and PIC.

To clarify how the alignment was optimized we have added to the supplement the info that the SLED was set to a low current for the alignment, ensuring the laser remained below threshold. And we added that maximizing the power of amplified spontaneous emission coupled through the PIC was used for finding an optimal alignment.

2. The GaN emitter has a relatively small mode size of $1.87 \times 0.6 \mu\text{m}$, most work of this type quantifies or models the optical loss at the coupling interface due to modal overlap with the spot size converter, and the associated coupling loss due to lateral or vertical misalignment in the two interfaces.

To model the expected optical loss and its sensitivity to transverse alignment we performed mode overlap calculations (using Lumerical). With the present design parameters we found good trade-off yielding a theoretical maximum mode overlap of 91% and an alignment tolerance of about 100 nanometer (details see Supplementary Information). Such values are well within the range of the three-axis stages used for chip-to-chip integration.

3. It might be of interest to the community to see the impact to optical loss in the alumina if it were not annealed at high temperature. Ion traps often have integrated metal electrodes for trapping of the ions, the presence of buried metals often precludes the use of high temperature process steps for annealing the oxides and alumina film. (or provide a reference to this

information)

Not annealing oxides at high temperatures is generally accepted as resulting in higher losses. The present process flow includes annealing the Al₂O₃ and 1 μm SiO₂ (high temperature LPCVD) top cladding to temperatures above 1100 °C [<https://doi.org/10.1364/opticaopen.25663656>], followed by deposition of the rest of the 8 μm total SiO₂ cladding (low temperature PECVD). The electrodes - in our case heater electrodes – are deposited afterwards, which would also be preferred for depositing the electrodes for an ion trap. Annealing after deposition of any metal electrodes at these temperatures might result in increased losses or contamination of the annealing chamber. Alternatively, buried electrodes can also be fabricated by etching trenches in the annealed Al₂O₃ and SiO₂ and subsequently depositing metal in the trenches. Afterwards, the trenches can be filled with low-temperature PECVD SiO₂.

We have added the info to the supplement that, similar to the laser tuning electrode, also the electrodes for an ion trap would be deposited in a final fabrication step after annealing.

Reviewer #3 (Remarks to the Author):

The manuscript presents a significant step forward in integrated photonics with the demonstration of a widely tunable and narrow-linewidth laser operating near the ultraviolet range. The use of aluminum oxide waveguide circuits and gallium nitride amplifiers is well-justified and innovative. The reported intrinsic linewidth of 300 kHz and the high output power achieved are particularly impressive and demonstrate the potential for enabling applications like quantum computing and optical clocks. This work introduces a laser that operates at the very edge of the UV spectrum, showing promise for extending photonics platforms into shorter wavelengths.

While the work is technically strong, the claims of deep-UV capability seem a bit overstated since the laser operates in the violet region, specifically 403–408 nm. The title and abstract could be adjusted to better align with the demonstrated results to avoid potential confusion for readers.

To avoid the impression of overstating claims (deep-UV capability) and to avoid potential confusion for readers, we followed the advice of the reviewer and placed modifications in the title, abstract and main text. The further vision of deep-UV capability of the approach is now left to the end of the discussion.

The novelty of the platform is evident, particularly in the use of aluminum oxide for UV-compatible feedback circuits, but the manuscript could benefit from a clearer distinction between incremental and groundbreaking contributions.

Following the reviewer's recommendation, we tried to better separate between incremental versus groundbreaking contributions. The only incremental parts are 1) using a known Vernier-based laser concept and 2) that we have limited ourselves to the shortest visible (violet) wavelength near the UV (405 nm) for which semiconductor optical amplifiers are presently commercially available.

The main novelties constituting a breakthrough towards future integrated UV lasers are 1) harnessing UV transparent materials (Al_2O_3 and SiO_2) in a novel, CMOS compatible fabrication 2) developing a full set of waveguide building blocks, 3) embedding them in complex and functional feedback circuits, and 4) bringing them to long-term stable laser operation via a UV-compatible hybrid integration technique. The high quality and completeness of these novelties, i.e., the full viability of the approach, is evidenced by record laser performance. This includes output power, coherence and tunability. Another evidence is that already the current laser reaches the requirements for use in combination with an integrated ion trap (here for 88-Sr^+ ions). The UV capability of the involved materials and integration technique opens the path towards chip integrated, highly coherent and tunable lasers in the UV, such as for portable optical clocks, for space applications, or as scalable light sources for ion-based quantum computing.

We addressed the above in the introduction (last paragraph) and throughout the discussion. Specifically to the discussion we added the following summarized statements:

‘We have shown the first chip-integrated extended cavity diode laser solely using UV transparent materials. Specifically, we use feedback waveguides made of aluminum oxide and silicon oxide. High performance of the laser is made possible through important advances such as: low losses, UV-compatible integration, optimized laser design and advanced laser control.’

The connection of this laser to quantum technology is promising, but the manuscript could elaborate further on how the laser's characteristics specifically meet the needs of quantum computing or other applications. The authors mention ion traps and cooling transitions, but to fully support the claims of usefulness in quantum applications, additional experimental validation would be very helpful.

We have carried out additional experiments serving to help the connection with application for an ion trap. As described in the revised supplemental information, the laser shows a high passive frequency stability when set to one of the fundamental transition frequencies of 88-Sr^+ ions (supplement Fig. SI-2c). As mentioned in an earlier response, during a 70+ minute recording the laser frequency stays mostly within the 22.1 MHz linewidth of the Sr-transition with minor, manual adjustments of a single thermo-optic phase shifter. The intrinsic linewidth (300 kHz) is about two-orders of magnitude below the linewidth of the transition, which would allow a straightforward active frequency stabilization and narrowing as with extended cavity diode lasers in bulk format. The output power is more than one order of magnitude higher than the typical hundred microwatt power required. The power surplus is beneficial for compensating experimental losses in integrated traps, such as due to the limited efficiency of surface grating couplers used for beam focusing.

For example, mode-hop-free tuning should be characterized to determine how far the laser can be tuned continuously, as this is a critical parameter for many quantum systems.

Indeed the capability of mode-hop-free tuning around and toward predefined, fixed frequencies is critical in many high-precision laser applications, including quantum applications. To check for this property, we have investigated mode-hop-free tuning of the laser across one of the two fundamental transition frequencies of 88-Sr^+ ions as marked by a high-accuracy wavelength meter at 734.99068 THz. These experimental results are now included in Fig. 2e. Separate measurements concluded that the maximum mode-hop-free tuning range is 21.5 GHz.

Even better, a demonstration of the laser being used in a real quantum experiment would significantly strengthen the claims. This could be as simple as a spectroscopy measurement that directly showcases the laser's applicability in a quantum setting. Without such demonstrations, the claims about potential applications risk appearing speculative and lack convincing evidence of practical usefulness.

We agree that an application of the laser in a real quantum experiment would be a great step, however, in reality such progress requires working in more steps. To underline that progressing in steps is required we have replaced the former Fig. 1a in the introduction (multiple ion traps fully integrated with lasers) by a more realistic view of a single trap partially still operated via fiber connections, presented as Fig. 3 (now in the discussion section).

To more detail, we expect that a next step following our mode-hop-free tuning demonstration is active frequency stabilization to ions in an existing bulk trap. As an intermediate solution, the laser output would be sent to the trap via fiber. Next, one may also design and fabricate separate Al_2O_3 -based lasers at different wavelengths, or even a multi-wavelength integrated source with a single feedback chip, however, using fibers for sending the output to an integrated trap would still be preferred. Directly diverting laser output to an integrated trap via Al_2O_3 waveguides would initially be done stepwise as well, such as we have indicated in Fig. 3 showing as an example two fiber coupled and two waveguide-coupled lasers. We note that in all this it remains of central importance that the Al_2O_3 waveguide platform leveraged here does not impose upfront transparency limitations, even for ions that require UV light.

It's a great piece of work, but it falls short in showing the laser in action within a real system. Including some system-level measurements would not only convince the audience of the laser's practicality but would also elevate the paper's impact significantly. For instance, integrating the laser into a testbed setup, such as for quantum spectroscopy or state preparation of ions, would add immense value. These demonstrations would provide tangible proof of the laser's capabilities and make the work far more compelling for both academic and industrial audiences.

We have shown mode-hop-free tuning, an essential feature required for these lasers to work in such systems. However, we think that the actual realization of such an ion trapping experiment is beyond the scope of this work, which is focused to lay the foundation for the required laser operation.

Overall, this work is a significant engineering achievement, and the authors have demonstrated strong expertise in photonics. The manuscript is well-written and presents solid results that will be of interest to the photonics community. Addressing the points mentioned above would strengthen the manuscript and make its contributions clearer to a broad audience. With some revisions and demonstrations of the laser's performance in a practical setting, this work has the potential to make a valuable and lasting impact on the field of integrated photonics and its applications in quantum technologies.

Review response letter NCOMMS-24-48081A

Our response to the reviewer questions is in blue.

Reviewer #1 (Remarks to the Author):

The current manuscript presents high-quality work that advances the state of the art in terms of technical performance.

We thank the reviewer for their positive words on our advancement of the state of the art.

The use of a UV-transparent platform is a noteworthy step; however, it appears to be an intermediate milestone rather than a definitive breakthrough toward the ultimate goal of realizing an integrated UV laser source. While the reported results are promising, they fall short of providing a scientifically rigorous demonstration that the proposed materials will indeed enable UV-integrated laser operation.

The reviewer argues with a lack of novelty solely based on a technical 3-nm detuning of the laser from what they desire it to be. We find such argument inappropriate because the future potential of a novel path does not hinge on a minor technical detail, but deserves discussion on a fundamental basis. Specifically, the reviewer inappropriately negates the UV capability of Al₂O₃ waveguides despite earlier UV demonstrations as singular passive components. Meanwhile these waveguides are shown to be low loss even down to 266 nm wavelength in the deep-UV [<https://doi.org/10.1117/12.3041427>]. Our waveguide circuit design and laser integration method is fully compatible with these UV wavelengths as well.

Furthermore, the manuscript lacks a concrete implementation in a real-world application where a UV laser is specifically required, which would strengthen the case for its practical relevance.

The most recent version includes a demonstration where we have aligned the laser frequency with extreme precision to within the natural linewidth of a Sr-transition, as measured using a wavelength meter. Indeed, the development of real-world integrated optical systems is in great need of integrated UV laser sources with exceptional frequency control, and this why we dedicated our research to present a laser that will meet this need. Only now, after the laser developments that we have presented, embedding such a laser in an ion trapping application can be taken up.

Additionally, the title remains misleading. It implies that widely tunable and narrow-linewidth violet lasers were achieved as a direct consequence of employing a UV-transparent material. This is inaccurate, as similar violet laser performance has already been demonstrated using silicon nitride—a material not classified as UV-transparent (see Ref. 30: doi: 10.1038/s41566-022-01120-w).

This statement is an obvious misinterpretation of facts, silicon nitride is fundamentally incapable to produce integrated lasers in the UV and thus has zero potential there. The named paper [Ref. 30] solely aims on the visible, and does it very well. In contrast, our approach is capable to open up the UV range, it exceeds all previous performance metrics by far and utilizes a different laser concept that provides precise frequency control (self-injection locking [30] vs extended cavity [this work], respectively).

In summary, the revised version does not adequately address the core concerns previously raised. As such, I believe the manuscript still falls short of the standards expected for publication in a multidisciplinary, high-impact journal.

We have received all comments, addressed them extensively, in this and previous response letters. Specifically, we are thankful for asking us to include a discussion of thermorefractive noise and autofluorescence in the revised manuscript.

Reviewer #2 (Remarks to the Author):

Thank you to the authors for satisfactorily addressing the main points I outlined in the previous draft. I have no further comments regarding the revision.

We thank the referee for their positive comment and questions that helped us improve our work.

Reviewer #3 (Remarks to the Author):

Thank you for your effort in addressing all the comments.

We thank the referee for comments in this process and for stating that all comments have been addressed.